# Predicting Solar Irradiance at Several Time Horizons Using Machine Learning Algorithms

**Chibuzor N. Obiora \*** , **Ali N. Hasan and Ahmed Ali**

Department of Electrical and Electronic Engineering, Faculty of Engineering and the Built Environment, University of Johannesburg, Johannesburg 2092, South Africa; alin@uj.ac.za (A.N.H.); aali@uj.ac.za (A.A.)
\* Correspondence: chibuzorobiora58@gmail.com

**Abstract:** Photovoltaic (PV) panels need to be exposed to sufficient solar radiation to produce the desired amount of electrical power. However, due to the stochastic nature of solar irradiance, smooth solar energy harvesting for power generation is challenging. Most of the available literature uses machine learning models trained with data gathered over a single time horizon from a location to forecast solar radiation. This study uses eight machine learning models trained with data gathered at various time horizons over two years in Limpopo, South Africa, to forecast solar irradiance. The goal was to study how the time intervals for forecasting the patterns of solar radiation affect the performance of the models in addition to determining their accuracy. The results of the experiments generally demonstrate that the models' accuracy decreases as the prediction horizons get longer. Predictions were made at 5, 10, 15, 30, and 60 min intervals. In general, the deep learning models outperformed the conventional machine learning models. The Convolutional Long Short-Term Memory (ConvLSTM) model achieved the best Root Mean Square Error (RMSE) of 7.43 at a 5 min interval. The Multilayer Perceptron (MLP) model, however, outperformed other models in most of the prediction intervals.

**Keywords:** deep learning; machine learning; solar irradiance; prediction; algorithms; time horizons

## 1. Introduction

Since the year of the combustion engine's inception, fossil fuels have been the primary energy source for most electricity-generating power plants worldwide [1]. They have been utilized for generating and supplying electricity in almost every sector, including transportation, telecommunication, hospitality, and housing. In addition to generating electricity, they are essential for powering several engines frequently employed to motorize power equipment for various industrial applications. It is impossible to overstate the dependence on non-renewable energy sources as the primary source of generating electricity, given that practically every sector of the global economy depends on access to electricity [2].

However, using fossil fuels seriously endangers the ecology since it contributes large quantities of carbon (IV) oxide emissions to the air. Carbon (IV) oxide, chlorofluorocarbons, and other pollutants are greenhouse gases (GHGs) that gradually increase the atmosphere's temperature and contribute to global warming. Global warming may gradually lead to the melting of the polar ice caps, increasing the sea's mean salinity [3]. In addition to contributing to global warming, fossil-fuel emissions and atmospheric moisture combine to cause acid rain, which is terrible for the ecosystem [4,5].

Recently, there has been a greater emphasis on research relating to estimating solar irradiance due to the ever-increasing demand for, and interest in, green energy [6]. It is crucial to accurately forecast solar irradiance when designing and managing photovoltaic (PV) systems [7,8]. For the power grid to run smoothly, or for the best control of the energy flows into the solar system, it is necessary to forecast the output power of solar systems. Concentrating on predicting solar irradiance before predicting the solar photovoltaic (PV) output is crucial [9].

The expanding demand for solar energy indicates that it is a viable option for most non-renewable energy sources. Since solar energy is a clean and abundant electric power source, it is preferred among all Renewable Energy Sources (RES).

Despite the recent increase in solar energy use in both the commercial and residential sectors, the intermittent behavior of solar energy's radiation intensity continues to pose a significant challenge to power grid operations. Most renewable energy sources suffer from unpredictability or an intermittent nature [10–12]. Due to the interplay between radiation and matter, solar irradiance is the radiative energy from the Sun that eventually reaches the photovoltaic (PV) cells [13,14]. The primary part of the Sun's rays extracted to produce electricity is solar irradiance, expressed in Watts/m$^2$.

In this study, to analyze Global Horizontal Irradiance (GHI) at different horizons, eight machine learning models were trained using data collected at different time horizons from Limpopo, South Africa. Machine learning models are used to predict the patterns of solar irradiance because of their ability to obtain hierarchical information from spatial–temporal data domains with tens of millions of parameters to mitigate the problem of intermittence [15,16]. After the models' training, there will always be a need to measure the prediction error. Some of the vital error metrics are Mean Absolute Error (MAE), Root Mean Squared Error (RMSE), and Coefficient of determination (R$^2$). These metrics indicate the performance of the models after training. The first contribution of this work is using and comparing eight machine learning (ML) algorithms with two years of datasets collected at different horizons of 5 min, 10 min, 15 min, 30 min, and 60 min from Limpopo. The results (outputs) were analyzed to determine the most accurate method. Another contribution of this work is determining the best horizon for accurately predicting solar irradiance using ML models. Unlike many related works, this study focused on predicting and assessing solar radiation patterns at five different time horizons. Most previous works considered only the accuracy of the Artificial Intelligence (AI) models at single time intervals, with no comparisons whatsoever with other possible time horizons with data collected from the same location. This study highlights the models' behavior and accuracy as the prediction intervals change. After training, the authors considered the models' accuracy with data gathered at 5 min, 10 min, 15 min, 30 min, and 1 h time horizons. Attention was also given to the effect of large volumes of data on the accuracy of deep learning models versus traditional machine learning models. It is clear from the literature review that dataset quality significantly influenced the accuracy of the forecasts, the locations where the data were collected, and the models used to make the predictions.

Contrary to this research, where the machine learning models were trained using data from several time horizons, many previous papers concentrated only on estimating solar irradiance without considering the time horizons of data collection. For example, in 2019, the authors of [17] predicted solar irradiance using data from a solar power plant in Dhaka, Bangladesh. Their findings revealed that an Artificial Neural Network (ANN) produced the best results with a coefficient of determination (R$^2$) = 0.999. In 2021, the authors of [18] carried out research to predict hourly solar irradiance using Extreme Gradient Boosting and Deep Neural Network models in Limberg, Belgium. Their most significant result is an RMSE of 51.35. Furthermore, the authors of [19] discovered that Extreme Gradient Boosting (XGB) outperformed Random Forest (RF) and ANN with a coefficient of determination (R$^2$) = 0.9996. They carried out this research in India, forecasting hourly solar irradiance based on input variables, including temperature, precipitation, aerosol data, sun angles, relative humidity, etc.

Furthermore, the authors of [20] predicted GHI in 2020 using the LSTM model and a training dataset from Goheung, Korea. The best result they achieved was an RMSE of 38.13 kW. In 2018, the authors of [21] estimated solar radiation using the Local Sensitive Hashing (LSH) model. They gathered their data from a solar power station in Ashland, Oregon, USA. The best performance that they reported was an RMSE of 4.23 kW. In addition, in 2016, the authors of [22] proved that Support Vector Machine (SVM) outperformed the Hidden Markov Model (HMM) in all weather situations in their experiment. Their objective

was to forecast Australia's short-term solar radiation for the next 5 to 30 min. Other papers surveyed [23–27] reviewed only forecasted GHI at a single time horizon.

The rest of this paper is organized as follows. The following section provides the materials and methods for predicting GHI at several intervals using machine learning algorithms. Analysis of the results in Section 3 focuses on the performance of each model after training with data collected at different time horizons. This section also includes a comparative assessment of the various models used. The concluding section of this study highlights the importance and contribution of this paper.

## 2. Materials and Methods

The experiment's data source was Solcast's Limpopo weather repository from 2020 to 2021 [28]. Five distinct time intervals were used to collect the data: 5, 10, 15, 30, and 60 min. After being cleaned and prepared, the data were ready for training the models. The complete data were divided into training and testing datasets after pre-processing. Eighty percent of the total data are contained in the training datasets. Only historical solar radiation was employed as an input parameter because the models used are time series. The epochs and batch size for fitting the deep learning models were set to 100 and 32, respectively. Each Deep Learning (DL) model underwent three or four timestep training rounds. This process indicates that data collected sequentially at specific time intervals for training at various layers will be provided as input. To reduce model overfitting and produce the best outcomes, the Adam optimizer and Rectified Linear Unit (ReLU) activation function were also used for the deep learning models. The overall procedure or methodology for training the models using input data collected at five different intervals is shown in Figure 1.

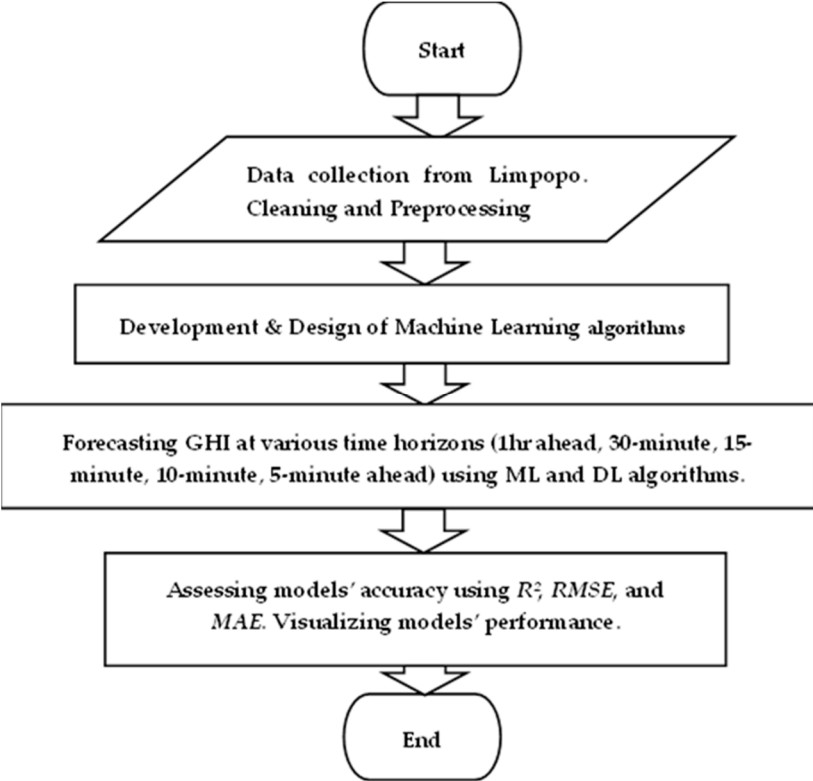

**Figure 1.** An illustration of the general methodology for machine learning-based GHI prediction at various time horizons.

The general procedure implemented in forecasting solar irradiance is summarized as follows: Collect and sort the data based on different time horizons, change the original data format to CSV files, cleanse the data by removing anomalies such as data recorded at nighttime hours; scale and normalize the data, split data into training and testing datasets,

design the deep learning network to train the data, train and test the data at several epochs, cross-validate the data, evaluate the performance of models by computing error rates and to apply the error metrics such as MAE and RMSE. Finally, visualize the performance of the models by generating graphs using the Seaborn libraries and Matplotlib.

Brief descriptions of the models used for solar irradiance forecasting in this paper are given below.

A.    Bidirectional Long Short-Term Memory (BiLSTM) network

The LSTM model learns a function that transforms a set of previous observations into a new observation [29]. It can be advantageous to let the LSTM model learn the input sequence both forward and backward and concatenate both interpretations for particular sequence-prediction problems. The Bidirectional LSTM (BiLSTM) model is what is used in this case. By enclosing the first hidden layer with the Bidirectional wrapper layer, one may create a Bidirectional LSTM for univariate time-series forecasting.

B.    Convolutional Long Short-Term Memory (ConvLSTM) network

Convolutional reading of the input is incorporated by each LSTM unit in ConvLSTM, causing convolutions to be performed by its gates. The ConvLSTM model can be used to forecast time series using a single variable, despite being designed to read two-dimensional spatial–temporal data. Since the layer needs input in the form of a collection of two-dimensional images, the data must be in the format [samples, timesteps, rows, columns, and features]. Each sample can be divided into smaller parts using subsequences, where timesteps represent the overall number of subsequences and columns represent the overall number of time steps for all subsequences.

C.    Convolutional Neural Network (CNN) time-series model

CNN is a multilayer perceptron model inspired by the workings of the brain. In tasks involving image processing and recognition, they have demonstrated dependability. CNNs were initially created to represent 2D image data but may now be used to model problems involving time-series prediction. Instead of seeing the input as having temporal steps, CNN sees it as a sequence over which convolutional read operations can be applied, much like a 1D image [30]. A 1D CNN is a CNN model with a hidden convolutional layer that operates on a one-dimensional sequence [31].

D.    CNN-LSTM hybrid model

A CNN model can be used in a hybrid model with an LSTM backend to interpret input subsequences collectively supplied to the LSTM model as a sequence. The name of this hybrid model is CNN-LSTM. The input sequences must first be divided into manageable pieces for the CNN model. For instance, univariate time-series data can be used to generate samples for input and output. It can take up to four steps as inputs and up to one step as an output. Then, each sample can be divided into two smaller samples for four timesteps.

E.    Multilayer Perceptron (MLP)

The concepts of backpropagation and hidden layers gave rise to Multilayer Perceptrons (MLPs). The weights are repeatedly adjusted using the backpropagation approach to reduce the difference between the observed output and the desired result. Because of the hidden layers, composed of neuron nodes positioned between input and output, neural networks can learn more complicated features. An MLP can therefore be thought of as a deep artificial neural network. It consists of several perceptrons and an input layer that receives signals.

F.    Random Forest (RF) model

Random forests, a machine learning method that uses the ensemble method, are also known as random decision forests. They are employed in classification and regression problems where building a large number of decision trees is a necessary part of the training process. Random decision forests were built to address the over-fitting issue affecting decision trees. The random subspace method was used to create the original random forest

algorithm [32]. Using the random subspace method, the stochastic discrimination strategy for classification is applied.

G.    Extreme Gradient Boosting (XGboost) Ensemble model

The stochastic gradient boosting machine learning technique is effectively followed by extreme gradient boosting, also referred to as XGBoost. Gradient boosting is one of the best techniques for creating predictive models. Gradient-boosted decision trees are used in the ensemble learning method XGBoost to improve performance and speed. Scalability is the primary driver of XGBoost's popularity in all situations. On a single machine, the model performs ten times quicker than currently used methods, and in distributed or memory-constrained environments, it is scalable to billions of samples. The objective of the design of XGBoost is to improve the amount of computational power available for boosted tree algorithms.

H.    Linear Regression (LR) Model

Typically, linear regression aims to fit a regression line to the data so that the divergence-related error is as small as possible. The slope coefficient (m) in linear regression is defined as the expected change in Y for a one-unit increase in X. The intercept parameter, bias b, is defined as the expected value of Y when X = 0. To minimize this deviation, we initially created an optimization problem by combining all the squared vertex deviations between the data points and regression lines. Then, to minimize the error function, the bias term (b) and vector (W) is selected [33]. In other words, simple linear regression aims to reduce the distance between the data points and the regression line.

*2.1. Performance Evaluation*

The error metrics employed in this work to evaluate the models' performance following training at each time horizon are briefly described in the following subsection.

2.1.1. Root Mean Square Error (RMSE)

Mean Squared Error (MSE) measures how well a regression line fits a group of data points. It accomplishes this by squaring the separations between the points and the regression line. It also highlights significant distinctions. The sum of a set of errors is the mean square error. The forecast is more accurate when the MSE value is lower. The effectiveness and error rate of any machine learning technique used to solve a regression problem can be assessed using the Mean Square Error (MSE). The method for computing the MSE and RMSE is shown in Equations (1) and (2).

$$\text{MSE} = (1/\text{N})\sum\nolimits_{t}(\text{A} - \text{F})^2 \tag{1}$$

Root Mean Square Error, RMSE $= \sqrt{\text{MSE}}$

$$\text{Normalized Root Mean Squared Error (NRMSE)} = \sqrt{\text{MSE}}/\rho \tag{2}$$

$\rho$ = mean of observed or actual values
$\rho = (\text{A}_{max} - \text{A}_{min})$;
A = actual data;
F = forecast data;
N = numberofobservations.

2.1.2. Mean Absolute Error (MAE)

The simplest metric for assessing prediction accuracy is the Mean Absolute Error (MAE). The mean absolute error is the average of all the absolute errors. The level of measurement accuracy is referred to as absolute error. The absolute error is the difference

between the actual and predicted data, expressed as an absolute number. Equation (3) demonstrates how to determine the MAE.

$$\text{MAE} = (1/\text{N})\sum_t |\text{A} - \text{F}| \tag{3}$$

### 2.1.3. Coefficient of Determination ($R^2$)

The coefficient of determination gives a percentage representation of how much variance within the y-values the regression model can represent. As a result, a value closer to 100% indicates a good fit, whereas a value near 0% indicates the opposite. The procedure for determining the coefficient of determination is expressed by a mathematical Equation (4).

$$R^2 = 1 - \frac{\text{RSS}}{\text{TSS}} \tag{4}$$

RSS = Residual sum of squares.
TSS = total sum of squares.

### 3. Results

Solcast [28] weather reports provided the data collected from Limpopo from the year 2020 to 2021. The entire data were divided into training and testing datasets. The training dataset contains 80% of the total data. The model's Mean Square Error (MSE) loss was plotted against epochs during training. All the deep learning models were trained for 100 epochs. As an example, Figure 2 depicts the performance of the Multilayer Perceptron (MLP) model at 5 min, 10 min, 15 min, 30 min, and 60 min prediction intervals.

The graphs in Figure 2 show that the MSE loss in each model during training was very high in the early epochs but drastically reduced after a few epochs. This indicates that the model was learning patterns in the data and improving its predictive ability. The MSE loss decreases as the number of epochs increases, eventually maintaining near-constant values until the final epochs. Figure 3 shows sample regression plots illustrating how well the data fit the MLP model. The regression lines with the closest data points perform better than those with data points further away.

A closer examination of the regression plots in Figure 3 reveals that they are distinct, despite their resemblance. The closer the data points are to the regression line, the better fit they are. Because the data points clustered more closely along the regression line, the graph in Figure 3a, which represents the performance of the MLP model trained with data collected at 5 min intervals, produced the highest accuracy. Because the data points in Figure 3e are further away from the regression line, the accuracy is considered lower. We plotted several graphs that show the differences or errors between the actual or measured solar radiation and the forecasted solar radiation in relationship with time to better illustrate how well the models performed. Figure 4 displays the graphs produced by the Random Forest (RF) model after training using data gathered at intervals of five, ten, fifteen, thirty, and sixty minutes.

The graphs show that the accuracy of the models decreased as the intervals at which the training data were collected widened. For example, actual and predicted solar irradiance values nearly overlapped when training data were collected at 5 min intervals while the differences between the actual and predicted values are visible in the graph at 60 min intervals. The following subsections present the models' results trained with data collected at various time horizons.

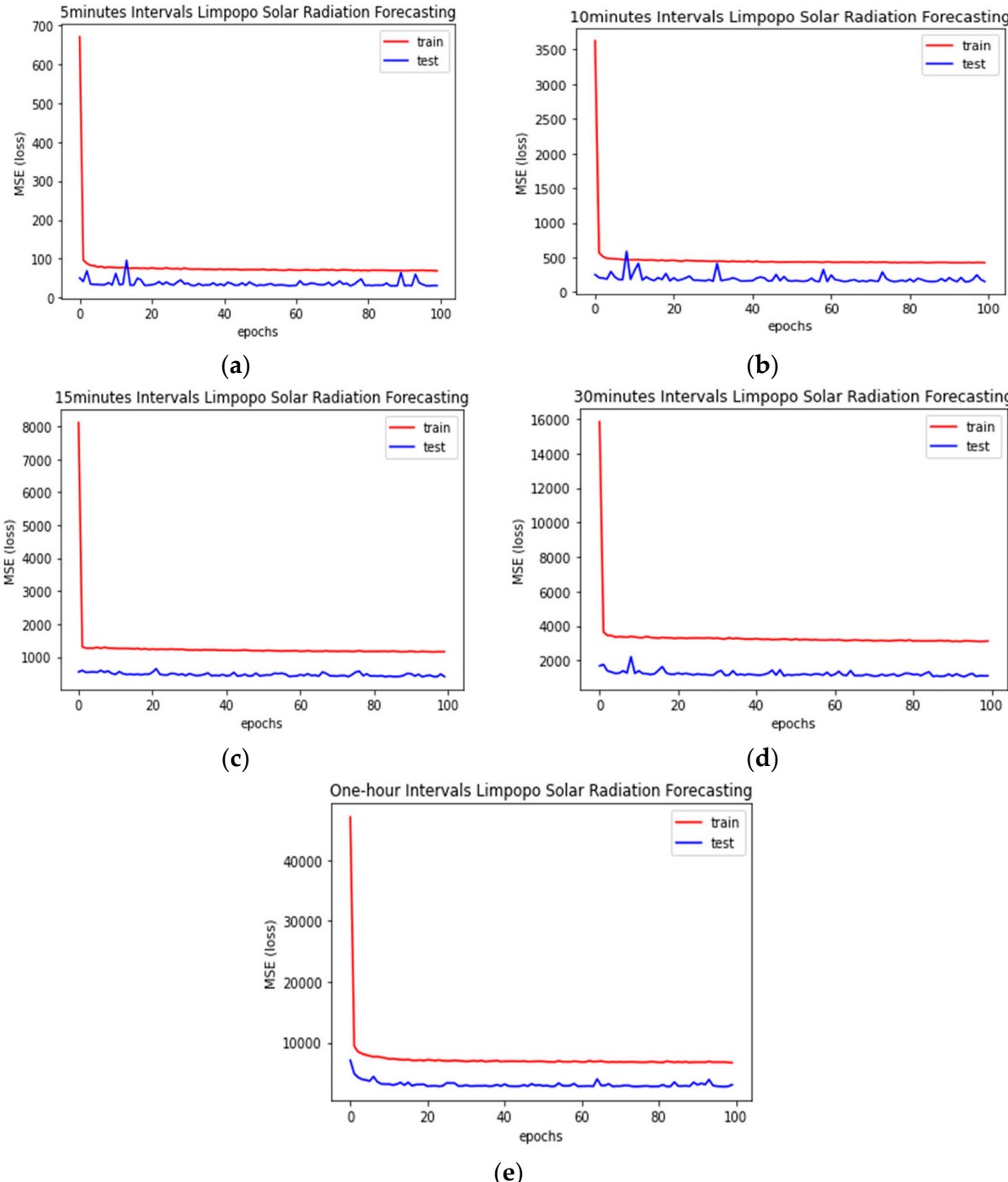

**Figure 2.** MSE loss versus epochs graphs. The MLP model's train–test loss graphs were obtained during training and validation, with data collected at (**a**) 5 min intervals, (**b**) 10 min intervals, (**c**) 15 min intervals, (**d**) 30 min intervals, and (**e**) 1 h intervals. In all prediction horizons, the MSE loss of the model was very high at the start of the training but dropped sharply after a few epochs to maintain a near-constant error rate until the final epochs.

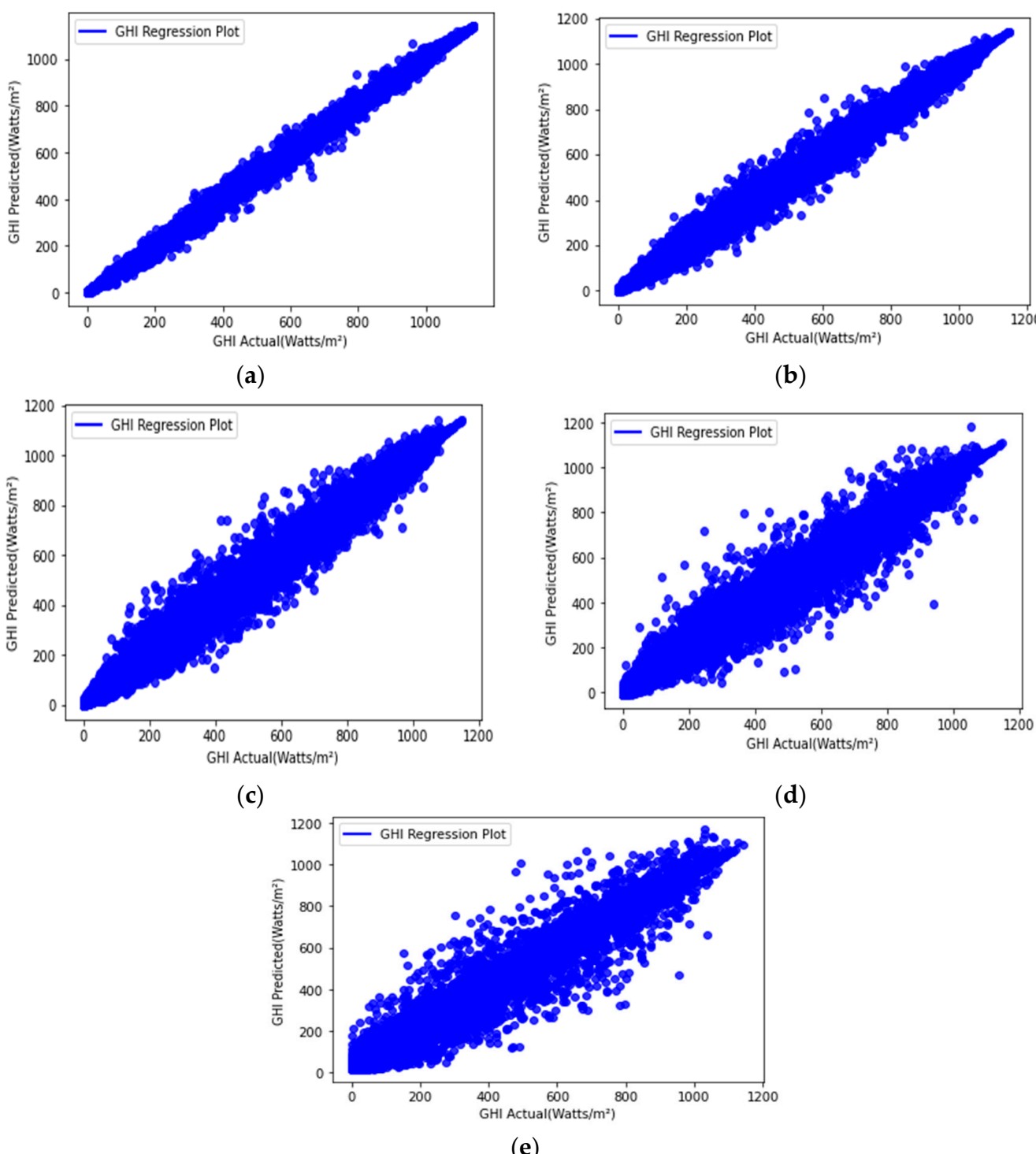

**Figure 3.** The MLP model's regression plots generated after training with two years of data collected at (**a**) 5 min horizon, (**b**) 10 min horizon, (**c**) 15 min horizon, (**d**) 30 min horizon, and (**e**) 1 h horizon. In all cases, the closer the data points are to the regression line, the higher the accuracy of the model.

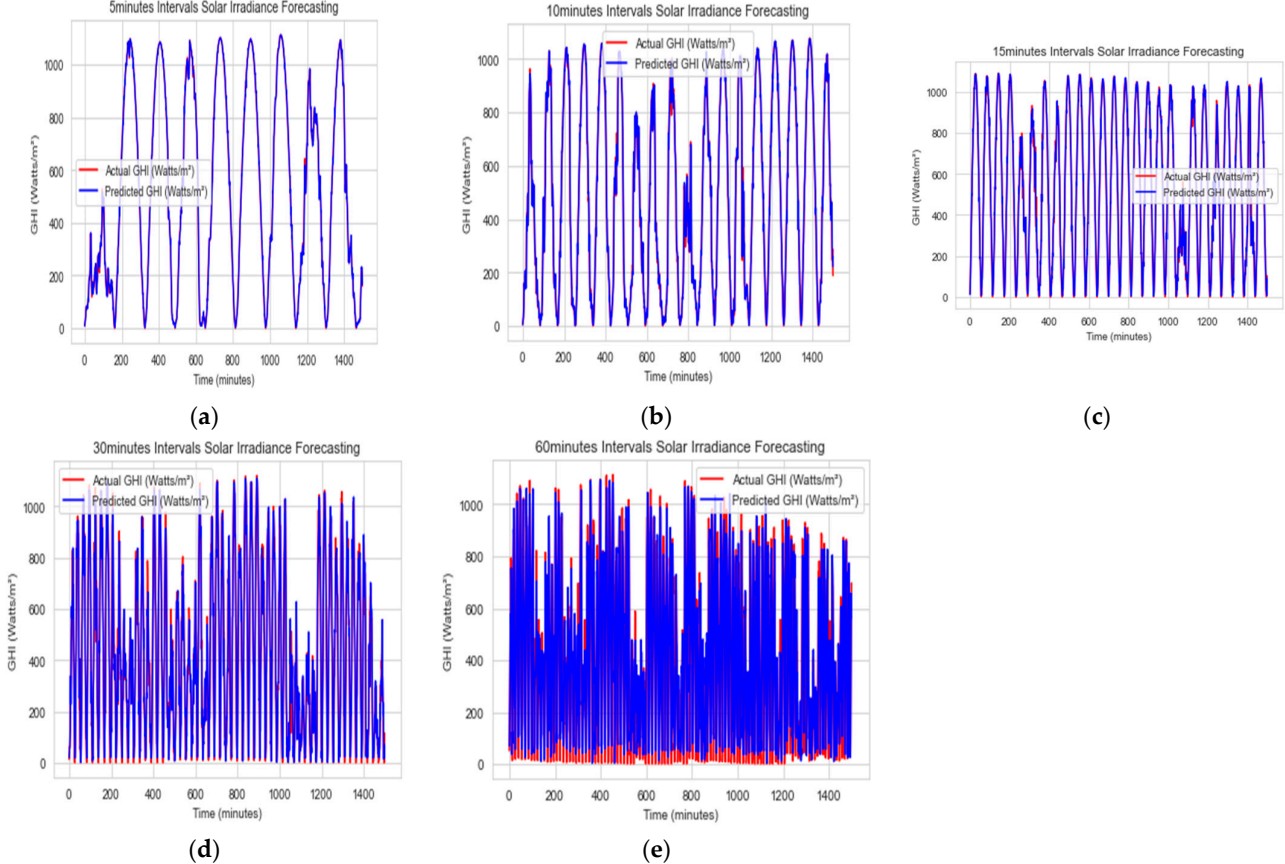

**Figure 4.** Plots of the RF model for training data gathered in Limpopo showing the predicted GHI versus the actual GHI at (**a**) 5 min, (**b**) 10 min, (**c**) 15 min, (**d**) 30 min, and (**e**) 60 min prediction intervals.

### 3.1. Results Obtained from Models Trained with Data Collected from Limpopo in Two Years at 5 min Intervals

The weather information was gathered between the years 2020 and 2021. The data were divided into training and testing datasets. The training dataset contains 80% of the data collected over two years. The total number of observed Global Horizontal Irradiance (GHI) collected at 5 min intervals is 105,045, with a mean of 448.54. As shown in Table 1, the ConvLSTM model outperformed the other models using these training data.

**Table 1.** The error metrics scores after training all the models with 80% of the data collected at a 5 min time horizon in Limpopo over a two-year period.

| Model | Error Metrics at 5 min Intervals | | | |
|---|---|---|---|---|
| | MAE | $R^2$ | RMSE | NRMSE (%) |
| BiLSTM | 3.82 | 0.9993 | 8.33 | 1.86 |
| CNN-LSTM | 9.99 | 0.9969 | 15.39 | 3.46 |
| ConvLSTM | 3.23 | 0.9994 | 7.43 | 1.66 |
| MLP | 3.35 | 0.9993 | 7.60 | 1.69 |
| CNN | 5.72 | 0.9989 | 10.23 | 2.28 |
| RF | 10.14 | 0.9986 | 13.97 | 3.11 |
| XGBoost | 10.08 | 0.9987 | 13.47 | 3.00 |
| LR | 14.54 | 0.9975 | 18.63 | 4.15 |

The RMSE value from the ConvLSTM model was the lowest, at 7.43. This was the best result possible from this dataset after 100 epochs. The MLP model was closer in performance to the CovLSTM model, scoring an RMSE of 7.60. Overall, the deep learning models produced fewer errors than the traditional machine learning models. The best-performing model outside the deep learning category was the XGBoost ensemble model, which had an RMSE of 13.47. The RF model, at 13.97, recorded almost the same accuracy as the XGBoost model. Figure 5 shows the boxplots for visualizing the prediction errors of the models after training with data collected at 5 min intervals. The plots reflect the results in Table 1.

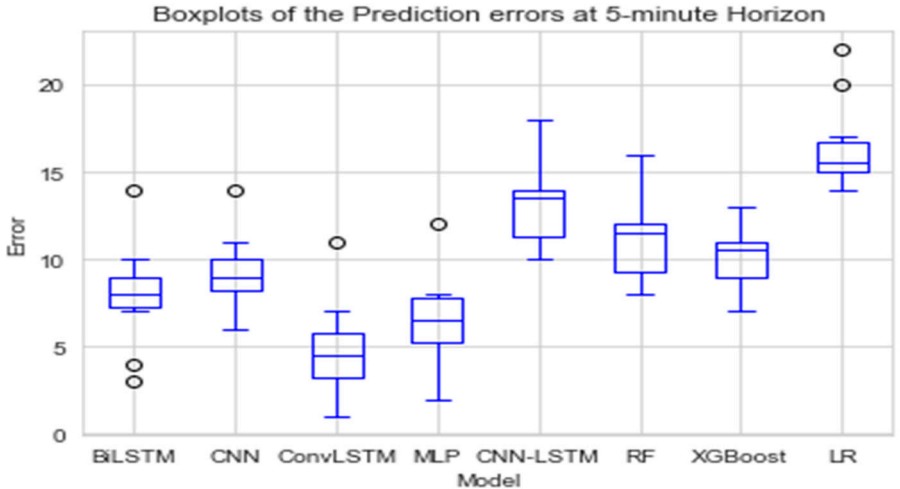

**Figure 5.** Boxplots showing the prediction errors of the models trained with data collected at a 5 min horizon.

*3.2. Results Obtained from the Models Trained with Data Collected from Limpopo in Two Years at 10 min Intervals*

Datasets for training and testing were generated from the entire data. Eighty percent of the data gathered over two years at 10 min intervals formed the training dataset. The total number of observations is 53,323, and the average or mean GHI measured by the observations is 418.41. As demonstrated in Table 2, the MLP model performed better using these data than the other machine learning models.

**Table 2.** The error metrics scores after training all the models with 80% of the data collected at a 10 min time horizon in Limpopo over a two-year period.

| Model | Error Metrics at the 10 min Horizon | | | |
|---|---|---|---|---|
| | MAE | $R^2$ | RMSE | NRMSE (%) |
| BiLSTM | 11.72 | 0.9957 | 20.36 | 4.87 |
| CNN-LSTM | 18.86 | 0.9898 | 31.99 | 7.71 |
| ConvLSTM | 12.47 | 0.9958 | 19.46 | 4.65 |
| MLP | 10.83 | 0.9961 | 18.87 | 4.51 |
| CNN | 13.65 | 0.9942 | 22.66 | 5.41 |
| RF | 19.98 | 0.9927 | 27.56 | 6.59 |
| XGBoost | 20.77 | 0.9928 | 27.26 | 6.52 |
| LR | 29.03 | 0.9866 | 37.29 | 8.91 |

The RMSE value obtained from the MLP model, 18.87, indicates the least error. This was the best result possible from this set of data. After training for 100 epochs, most of

the Deep Learning models performed well. The ConvLSTM model was slightly behind the MLP model, having recorded an RMSE value of 19.46. The BiLSTM and CNN models reported 20.36 and 22.66, respectively. Among all the models, the Linear Regression (LR) model produced the worst RMSE score of 37.29 in this forecasting time interval. The difference in the results produced by the XGBoost and RF models is small. The XGBoost scored 27.26, while the RF model recorded an RMSE of 27.56. The boxplots in Figure 6 show the prediction errors of all the models after training with data collected at a 10 min horizon. The plotted graphs reflect the results in Table 2.

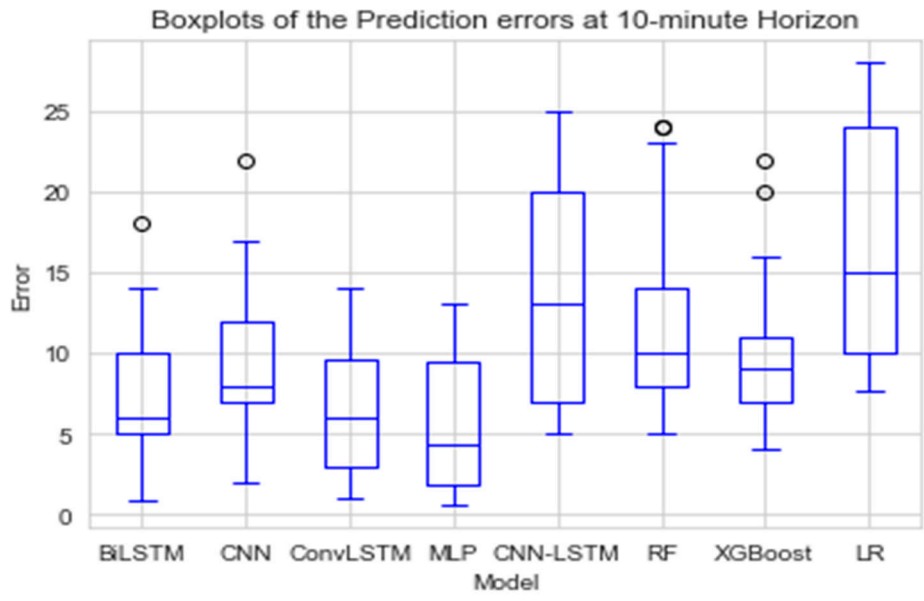

**Figure 6.** Boxplots of the prediction errors of the models after training with data collected at a 10 min horizon.

*3.3. Results Obtained Using Models Trained on Limpopo Data Collected over Two Years at a 15 min Horizon*

The data were divided into training and testing datasets. The training dataset contains 80% of the two years of Limpopo data collected at 15 min intervals. The observed or actual number of GHI is 35,982, with a mean of 414.58. Table 3 shows that the MLP model outperformed the other machine learning models after training on these data.

**Table 3.** The error metrics after training the models with 80% of the data collected at 15 min intervals in Limpopo over a two-year period.

| Model | Error Metrics at 15 min Intervals | | | |
|---|---|---|---|---|
| | MAE | $R^2$ | RMSE | NRMSE (%) |
| BiLSTM | 20.75 | 0.9886 | 34.37 | 8.29 |
| CNN-LSTM | 28.6 | 0.9784 | 43.01 | 10.31 |
| ConvLSTM | 19.94 | 0.9887 | 32.12 | 7.77 |
| MLP | 17.85 | 0.9892 | 31.38 | 7.57 |
| CNN | 20.39 | 0.9879 | 33.43 | 8.05 |
| RF | 28.37 | 0.9849 | 39.12 | 9.44 |
| XGBoost | 29.02 | 0.9851 | 38.87 | 9.38 |
| LR | 40.46 | 0.9734 | 51.88 | 12.51 |

The RMSE value obtained from the MLP model was the lowest, at 31.38. This was the best result possible from this dataset after training up to 100 epochs. The ConvLSTM model recorded 32.12 as its RMSE to place second in terms of performance. The CNN and BiLSTM models took the third and fourth positions, respectively. The XGBoost model was next in line with an RMSE of 36.87. The XGBoost model slightly performed better than the RF model, which registered an RMSE value of 39.12. Again, the LR model produced the worst results, with an RMSE of 51.88. Figure 7 shows the boxplots of the models for the prediction errors recorded after training with data collected at a 15 min horizon. The boxplots reflect the results in Table 3.

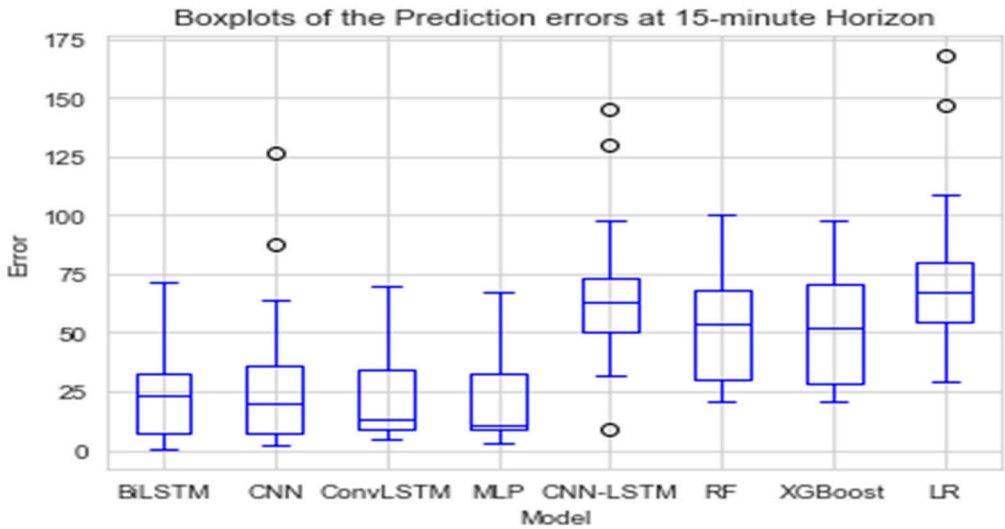

**Figure 7.** Boxplots of the prediction errors of the models after training with data collected at a 15 min horizon.

### 3.4. Results Obtained Using Models Trained on Limpopo Data Collected over Two Years at a 30 min Horizon

The data were divided into training and testing datasets. The training dataset contains 80% of the total data. The data collection at the 30 min horizon has 18,201 observations, and the mean of the actual or observed data is 408.81. As shown in Table 4, the ConvLSTM model outperformed the other machine learning models after training using these data.

**Table 4.** The error metrics after training the models with 80% of two years of data collected at 30 min intervals.

| Model | Error Metrics at 30 min Intervals | | | |
|---|---|---|---|---|
| | MAE | $R^2$ | RMSE | NRMSE (%) |
| BiLSTM | 32.62 | 0.9711 | 55.71 | 13.63 |
| CNN-LSTM | 48.96 | 0.9436 | 69.95 | 17.16 |
| ConvLSTM | 33.13 | 0.9714 | 51.18 | 12.52 |
| MLP | 33.16 | 0.9704 | 52.99 | 12.96 |
| CNN | 32.88 | 0.9697 | 53.19 | 13.01 |
| RF | 47.86 | 0.9613 | 64.82 | 15.86 |
| XGBoost | 49.03 | 0.9617 | 64.52 | 15.78 |
| LR | 70.36 | 0.9298 | 87.32 | 21.36 |

The RMSE value obtained from the ConvLSTM model was the lowest, at 51.18. This was the best result from this training dataset. The performance pattern of the models at

30 min prediction intervals is similar to the one at a 15 min interval, except that the MLP and ConvLSTM models swapped positions. Here, the MLP model took the second spot as it recorded an RMSE value of 52.99. With an RMSE of 87.32, the LR consistently produced the worst results. Figure 8 shows the boxplots of the models for the prediction errors recorded after training with data collected at a 30 min horizon. The plots reflect the results reported in Table 4.

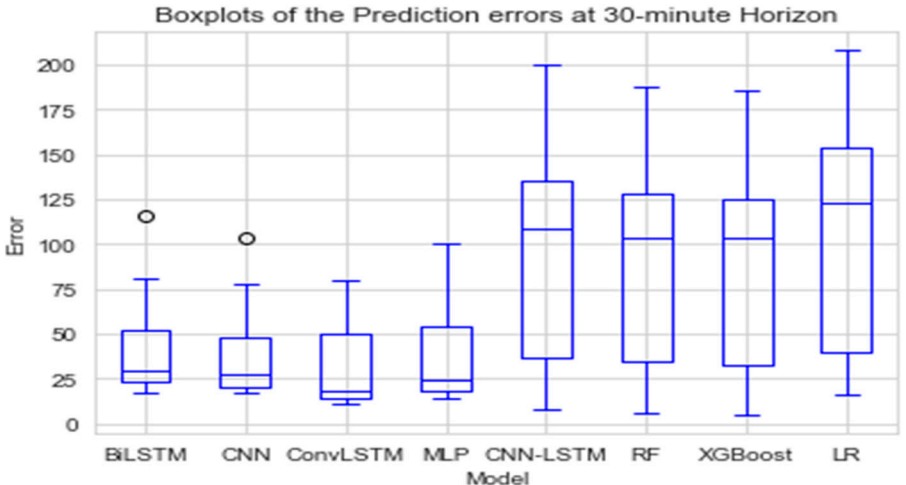

**Figure 8.** Boxplots of the prediction errors of the models after training with data collected at a 30 min horizon.

### 3.5. Results from the Models Trained with Data Collected from Limpopo in Two Years at a 60 min Horizon

The data were divided into training and testing datasets. The training dataset contains 80% of the total data. The total number of observations for the data collected at the 60 min horizon was 9373, with a mean of 394.11 for the observed or actual GHI. The MLP model outperformed the other machine learning models using the training dataset after 100 epochs, as shown in Table 5.

**Table 5.** The error metrics after training the models with 80% of two years of data collected at 60 min intervals from Limpopo.

| Model | Error Metrics at the 60 min Horizon | | | |
|---|---|---|---|---|
| | MAE | $R^2$ | RMSE | NRMSE (%) |
| BiLSTM | 51.48 | 0.9369 | 76.11 | 19.31 |
| CNN-LSTM | 74.47 | 0.8901 | 97.59 | 24.77 |
| ConvLSTM | 55.75 | 0.9307 | 79.98 | 20.21 |
| MLP | 51.56 | 0.9355 | 75.52 | 19.15 |
| CNN | 51.62 | 0.9361 | 76.8 | 19.49 |
| RF | 80.27 | 0.8897 | 107.07 | 27.16 |
| XGBoost | 84.5 | 0.8871 | 108.31 | 27.48 |
| LR | 121.46 | 0.7931 | 146.62 | 37.20 |

The RMSE value obtained from the MLP model was the lowest, at 75.52. After 100 epochs, this was the best result obtained using this training dataset. The BiLSTM model scored the second-best result with an RMSE of 76.11. The CNN model scored 76.8, while the ConvLSTM model scored 79.98 to take the third and fourth positions, respectively. The LR model produced the worst RMSE of 146.62 once more. For more illustration, Figure 9

shows the boxplots of the models for the prediction errors obtained after training with data collected at the 60 min horizon. The plots of the individual models are a reflection of the results in Table 5.

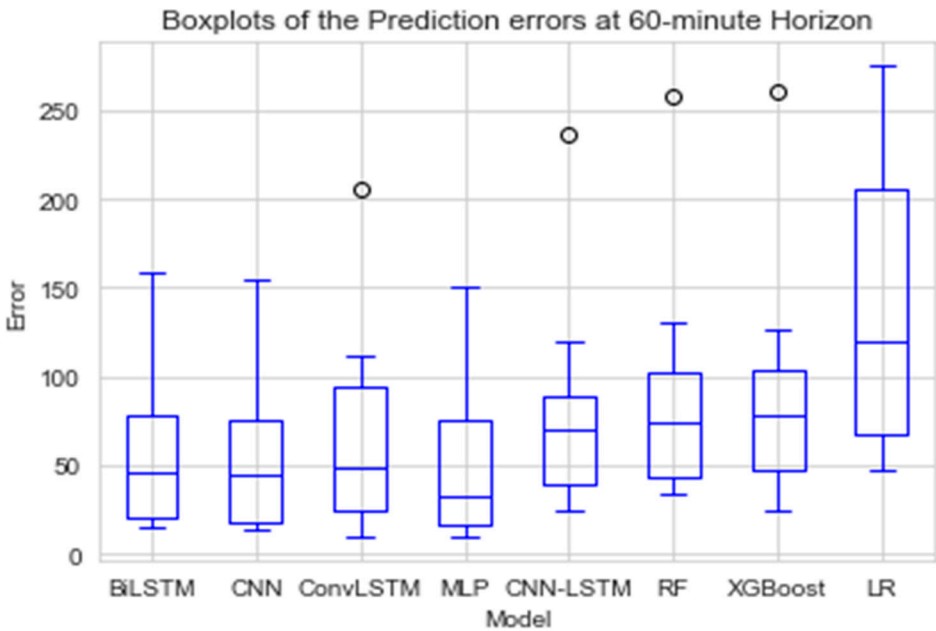

**Figure 9.** Boxplots of the prediction errors of the models after training with data collected at the 60 min horizon.

Figure 9 shows the MLP model had the lowest prediction error of 75.52. The LR model produced the worst result, with an RMSE of 146.62.

With training data collected at three of the five horizons, the MLP model produced the fewest errors after training. The ConvLSTM model was the best-performing in two other experiments, which trained with data collected at 5 min and 30 min intervals. Other models, such as the BiLSTM and CNN, performed creditably but were slightly outperformed by the ConvLSTM and MLP models. Table 6 summarizes the outcomes of predictions made using artificial intelligence-based models over various time horizons.

**Table 6.** A summary of error metrics obtained from experimenting with data collected at various time horizons.

| Model | RMSE Scores Obtained after Training at Various Time Horizons | | | | |
| --- | --- | --- | --- | --- | --- |
| | 5 min | 10 min | 15 min | 30 min | 60 min |
| BiLSTM | 8.33 | 20.36 | 34.37 | 55.71 | 76.11 |
| CNN-LSTM | 15.39 | 31.99 | 43.01 | 69.95 | 97.59 |
| ConvLSTM | 7.43 | 19.46 | 32.12 | 51.18 | 79.98 |
| MLP | 7.6 | 18.87 | 31.38 | 52.99 | 75.52 |
| CNN | 10.23 | 22.66 | 33.43 | 53.19 | 76.8 |
| RF | 13.97 | 27.56 | 39.12 | 64.82 | 107.07 |
| XGBoost | 13.47 | 27.26 | 38.87 | 64.52 | 108.31 |
| LR | 18.63 | 37.29 | 51.88 | 87.32 | 146.62 |

According to the pattern in Table 6, the RMSE scores generally increase as the time horizon over which the data were collected expands. For example, the Bidirectional LSTM (BiLSTM) produced an RMSE of 20.36 at 10 min prediction intervals and 34.37 at 15 min

intervals. This pattern holds across all the models used in this study. This indicates that the intermittence in solar irradiance becomes more unpredictable as the data collection intervals widen. The fluctuations in solar radiation are more gradual and tractable when data are collected over a shorter time horizon. The pattern of fluctuations in solar radiation may have completed a cycle of change by the time the data are collected at longer intervals, increasing the training errors produced by the models. Figure 10 shows a chart that visualizes the performance of the models after training with data collected at various time horizons.

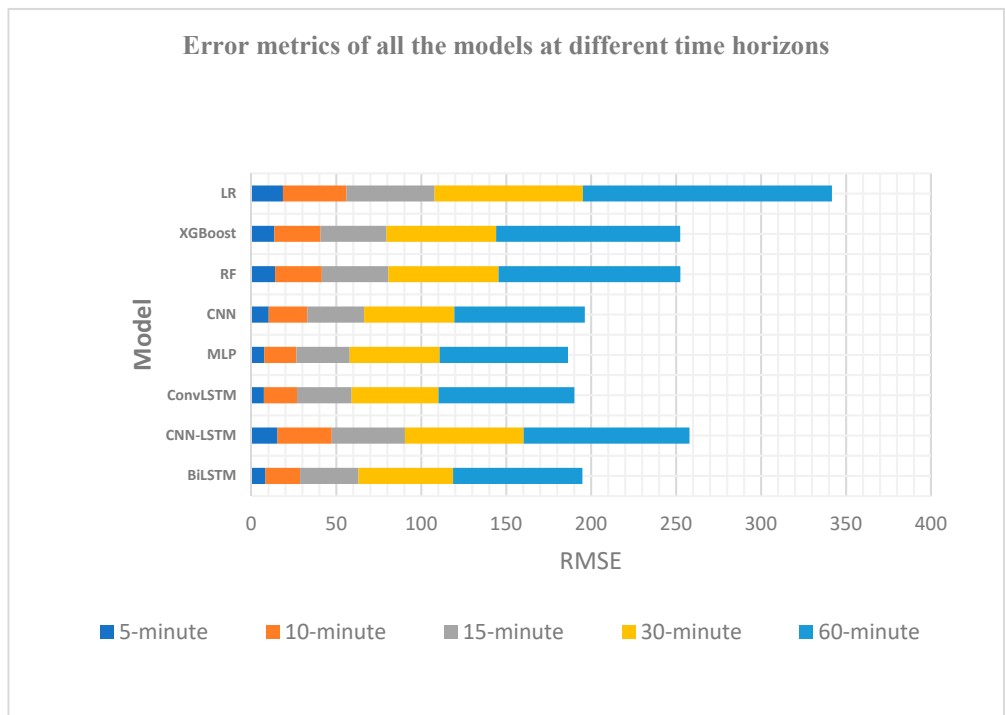

**Figure 10.** Chart displaying machine learning models' performance on Limpopo data at various prediction intervals.

According to the bar chart, the MLP model performed the best after training. The LR model produced the most significant errors in predictions. MLP > ConvLSTM > BiLSTM > CNN > XGBoost > RF > CNN-LSTM > LR is the order of performance of the models. From the result analysis, the MLP model can be considered the best model for tracking or predicting fluctuations in solar radiation with data collected at different time horizons in Limpopo.

## 4. Conclusions

Machine learning techniques are increasingly being used to forecast solar radiation. This drive is credited to their capacity to track solar irradiance patterns more precisely than traditional physical and statistical techniques. In this study, the authors predicted and analyzed solar radiation patterns over a range of time intervals using eight Machine Learning models. This study predicted the behavior of solar irradiance at various intervals, namely, five minutes, ten minutes, fifteen minutes, thirty minutes, and one hour, in contrast to many previous works evaluated, which concentrated on predicting solar radiation with data gathered at only one interval.

The findings demonstrate that when training data intervals widen, the accuracy of the models generally declines. This study's deep learning models outperformed more conventional machine learning models in accuracy. The efficiency of the deep learning models can be attributed to the fact that sizable quantities of data were required to completely capture Limpopo's two-year solar radiation trends. Deep learning models are known to

get more accurate as training data volumes increase. According to the findings, the MLP model performed better than other models in most predicted intervals. However, the RMSE produced from the ConvLSTM model at five-minute intervals was 7.43. This was the best result obtained during the experiments. However, the MLP model outperformed the other models at 10 min, 15 min, and 60 min predicted intervals. The performance of the MLP model was closely followed by the ConvLSTM model which produced the best results at 5 min and 30 min intervals. The BiLSTM and CNN models did very well but were slightly behind the MLP and ConvLSTM models, respectively. One factor that gave the MLP model an advantage was that it can achieve the same accuracy ratio even with smaller data samples. The authors believe that increasing the data size used in the experiment may likely push the other deep learning models to perform better than the MLP model. It is proposed that using the MLP, ConvLSTM, or BiLSTM models to forecast solar radiation in Limpopo solar plants would be beneficial in tracking its patterns and thus improving the reliability of solar electricity connected to the smart grid.

**Author Contributions:** Conceptualization, C.N.O., A.N.H. and A.A.; methodology, C.N.O.; software, C.N.O.; validation, C.N.O., A.N.H. and A.A.; formal analysis, C.N.O.; investigation, C.N.O., A.N.H. and A.A.; resources, C.N.O., A.N.H. and A.A.; data collection, C.N.O.; writing—original draft preparation, C.N.O.; writing—review and editing, C.N.O., A.N.H. and A.A.; visualization, C.N.O.; supervision, A.N.H. and A.A.; project administration, A.N.H. and A.A. All authors have read and agreed to the published version of the manuscript.

**Funding:** This research received no external funding.

**Institutional Review Board Statement:** Not applicable.

**Informed Consent Statement:** Not applicable.

**Data Availability Statement:** Data will be available on request.

**Acknowledgments:** The authors kindly thank the editor and reviewers who spent their valuable time improving the paper.

**Conflicts of Interest:** The authors declare no conflict of interest.

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
