# Peer review of "Predicting Solar Irradiance at Several Time Horizons Using Machine Learning Algorithms"

_sustainability, doi:10.3390/su15118927_

Round 1
Reviewer 1 Report
The authors of this paper concluded that the MLP, ConvLSTM, and BiLSTM models are the best for predicting solar radiation at solar facilities in Limpopo, which could increase the reliability of solar energy connected to the smart grid.
Can you provide additional information with figures and diagrams to demonstrate how your technique improves the reliability of solar integration into the smart grid?
Reviewer 2 Report
Dear Authors:
This paper investigates the use of machine learning models to forecast solar irradiance for power generation in Limpopo province, South Africa. Eight machine learning models were trained using data collected over various time horizons. The study found that the accuracy of the models decreases as the prediction horizons increase, and deep learning models generally outperform conventional machine learning models. The Convolutional Long Short-Term Memory (ConvLSTM) model achieved the best Root Mean Square Error (RMSE) at 5-minute intervals, while the Multilayer Perceptron (MLP) model outperformed other models in most prediction intervals. However, several major concerns are listed as follows.
1- Introduction: Please clearly highlight the contribution of this paper compared to the state-of-the-art methods.
2- The state of the art should be improved to show the benefit of the proposed approach. I recommend adding the following references,
a)https://doi.org/10.3390/sym12111830
b) https://doi.org/10.1016/j.enconman.2020.112582
c) 10.1109/INDIN45523.2021.9557405
d) https://doi.org/10.1016/j.renene.2016.12.095
3) Page 2, line 76, please avoid merging several references together. Please describe each reference separately, which is more helpful for the readers. For each reference, provide the developed approach, obtained results, advantages, and limitations if possible.
4- This paper presented an application of well-known machine learning and deep learning methods to forecast solar irradiance, and I cannot find any new contribution to the methodology. The contribution is missed in this paper. This part should be clarified.
5) Figure 1 should be improved.
6) The description of the investigated methods should be improved.
7) Equations 1, 2, 3, and 4 should be clearly presented.
8) The quality of figures must be improved.
9) Providing the values of the hyperparameters.
10) It is not clear how multi-steps forecasting is performed. Please clarify with equations.
11) The preprocessing step should be presented, particularly for the CNN-LSTM and the CNN models.
12) The authors present a comparison in the application part. However, in my opinion, this is not enough, they need to show more comparative work.
13) The paper reads as merely "we applied this model to that dataset and obtained those results", which does not contribute a great deal of knowledge. Consider a deeper analysis that discovers why each approach would be better.
The assessment of the forecasting accuracy should be improved.
14) Try to add statistical descriptive analysis of the used data
15) Please include the boxplot of the forecasting error for each model.
16) Authors should also conduct some statistical test, such as Diebold and Mariano (DM) test to ensure the superiority of the proposed approaches, i.e., how could authors ensure that the results of one model is statistically significant than the other?
17) Limitations and complexity of the proposed approach should be highlighted.
18) Minor grammar and syntax issues need correction, and writing norms need to be strengthened.
Reviewer 3 Report
the paper discusses a comparative study of different machine learning and deep learning models for the solar irradiance prediction problem.
the paper also has a new contribution which is working with dataset collected from different time horizons
the paper needs the following modifications though:
- the contribution of the paper needs to be highlighted (for me it sound like comparing different ML and DL models on a dataset and comparing results).
- the findings of the paper need to be highlighted also in my opinion more scientific conclusions can be concluded from results like the difference between these models in different time intervals simulations.
For example, it can be seen that linear regression has the worst RMSE for all simulations. similar conclusions will give the paper more scientific depth.
try to answer these questions. (Why CNN-LSTM is the worst type of LSTM variants in all experiments) I mean having the biggest RMSE. and so on.
some other small modifications: -
- when using multiple references don't put them like this [3],[4] it should be [3,4]
- you need a zoom in figure for figure 4 to show how the predicated value match with the real value.
- the model subsection numbering (most models have the subsection A.)
- you are using one year data is there a justification not to use more recent data?
- in line 54 you use the word power machine learning (please remove) what you mean by powerful.
- the last section in introduction ( the organization section needs to be rewritten
the section describing the XGBoost needs improvement.
- please add a notation table for equation symbols (what is N and what is At and so on)
Round 2
Reviewer 2 Report
Dear Authors,
Thank you for the revised manuscript. There are still some major comments to be carefully addressed.
1) The results in Figure 11 show that all models provide similar results. it is very strange. Please check carefully.
2) The quality of the figures should be improved.
3) The results displayed in Figures 6, 7, 8, and 9 are already reported in the tables. There is no need for these figures.
4) Please provide the boxplot of prediction errors obtained by each model for each case, i.e., 5minutes, 10minuntes,...
Regards,
Reviewer 3 Report
I thank the authors for addressing my comments
Round 3
Reviewer 2 Report
Dear Authors,
"In Figure 8, we observe that the prediction errors of RF, XGBoost, and LR are almost similar. However, their results in Table 4 are very different, particularly for LR. The same problem appears in Figure 9."
"Boxplots should be presented together in the same figure to facilitate clear comparisons. In each figure, approaches should be listed on the x-axis, and prediction errors should be placed on the y-axis."
A careful revision is needed to avoid any confusion.
Round 4
Reviewer 2 Report
Dear Authors:
While almost all major comments have been addressed, further revisions are necessary to improve the paper. Here are some comments that can be considered to improve this version.
1) Deeper discussion should be included when analyzing the results.
2) In the conclusion, try to replace "MLP > ConvLSTM > BiLSTM
> CNN > XGBoost > RF > CNN-LSTM > LR", by a clear sentence. Providing some discussion about why MPL is providing better results than deep learning models (ConvLSTM > BiLSTM, CNN, CNN-LSTM )?
3) The quality of the figures needs improvement.
